# Anti-Inflammatory Function of Plant-Derived Bioactive Peptides: A Review

**DOI:** 10.3390/foods11152361

**Published:** 2022-08-06

**Authors:** Wanlu Liu, Xinwei Chen, He Li, Jian Zhang, Jiulong An, Xinqi Liu

**Affiliations:** 1National Soybean Processing Industry Technology Innovation Center, Beijing Technology and Business University (BTBU), Beijing 100048, China; 2Beijing Advanced Innovation Center for Food Nutrition and Human Health, Beijing Technology and Business University (BTBU), Beijing 100048, China

**Keywords:** plant-derived bioactive peptides, anti-inflammatory mechanisms, structure–activity relationship, enzymatic processing

## Abstract

Inflammation is considered to be a crucial factor in the development of chronic diseases, eight of which were listed among the top ten causes of death worldwide in the World Health Organization’s World Health Statistics 2019. Moreover, traditional drugs for inflammation are often linked to undesirable side effects. As gentler alternatives to traditional anti-inflammatory drugs, plant-derived bioactive peptides have been shown to be effective interventions against various chronic diseases, including Alzheimer’s disease, cardiovascular disease and cancer. However, an adequate and systematic review of the structures and anti-inflammatory activities of plant-derived bioactive peptides has been lacking. This paper reviews the latest research on plant-derived anti-inflammatory peptides (PAPs), mainly including the specific regulatory mechanisms of PAPs; the structure–activity relationships of PAPs; and their enzymatic processing based on the structure–activity relationships. Moreover, current research problems for PAPs are discussed, such as the shallow exploration of mechanisms, enzymatic solution determination difficulty, low yield and unknown in vivo absorption and metabolism and proposed future research directions. This work aims to provide a reference for functional activity research, nutritional food development and the clinical applications of PAPs.

## 1. Introduction

Inflammation is a complex physiological response that protects the body from damage caused by external toxins and stimuli through multiple mechanisms [1]. It is the way in which the body heals itself after injury, repairs damaged tissue and fights off pathogens. Inflammation may be either acute or chronic depending on its duration [2]. The former is a rapid, early-stage response, usually lasting from a few minutes to a few days, which assists the organism fight against microbial infections and in wound healing. However, when an excessive inflammatory response progresses to the chronic stage, it can last for weeks or even years; and the onslaught of inflammatory factors leads to cell necrosis, decreased metabolism and immune function, eventually resulting in tissue damage and organ dysfunction. The risk of chronic diseases is increased when an organism is in a chronic state of suboptimal health. Studies have demonstrated that numerous chronic diseases are associated with inflammation, including aging [3], inflammatory bowel disease [4], cardiovascular disease [5], osteoporosis [6], cancer [7], obesity [8] and neurodegenerative diseases [9]. Therefore, damage from inflammation is one of the greatest health challenges of the 21st century and a leading cause of death globally, and an effective means to combat inflammatory damage is critical for the life and food sciences.

Non-steroidal alcohols (NSAIDs), such as aspirin and ibuprofen, are a class of chemically synthesized anti-inflammatory drugs that do not contain steroidal structures [10]. They are currently the most used anti-inflammatories worldwide; however, with the increased use of NSAIDs, unwanted side effects, such as gastrointestinal toxicity and kidney damage, are being increasingly observed [11]. In response, the anti-inflammatory functions of natural active ingredients, represented by plant-derived bioactive peptides, have attracted attention in recent years. Numerous studies have confirmed that plant-derived bioactive peptides can resist excessive inflammatory responses by modulating inflammatory signaling pathways and inhibiting the secretion of inflammatory factors [12,13,14]. Plant-derived proteins are widely available in cereals, nuts, fruits and other natural sources. Plant-derived bioactive peptides can be obtained in large amounts by enzymatic hydrolysis, chemical hydrolysis and microbial fermentation from plant-derived proteins. Plant-derived bioactive peptides are characterized by their low molecular weights and their easy digestion and absorption. Plant-derived bioactive peptides can be rapidly absorbed by the body through a specific transport mode in the intestines, differently from large protein molecules and even small amino acid molecules; they are transported to the target site to exert anti-inflammatory effects [12,15]. Their transit process minimizes energy consumption and reduces the burden on the gastrointestinal tract, which is ideal for those suffering from chronic diseases. Consequently, plant-derived bioactive peptides are receiving attention in both nutrition and pharmaceutical fields.

In this paper, the action mechanism of plant-derived anti-inflammatory peptides (PAPs), the factors affecting their anti-inflammatory effects and the effects of enzyme types on the processing of PAPs are reviewed. The relationship between the structural composition of plant-derived bioactive peptides and their anti-inflammatory activity is summarized in detail. Furthermore, the current research limitations and future research surrounding PAPs are analyzed to provide a reference for their development and applications.

## 2. Mechanism of the Anti-Inflammatory Effects of Plant-Derived Bioactive Peptides

Inflammation occurs after inflammatory pathways are stimulated by inducing factors and inflammatory factors are released [16]. Inducing factors include lipopolysaccharide (LPS), dextran sodium sulphate (DSS) and other toxicants. Inflammation is primarily mediated via the mitogen-activated protein kinase (MAPK) pathways, the nuclear factor kappa B (NF-κB) pathway and the phosphatidylinositol 3-kinase/protein kinase B (PI3K/Akt) pathway. The action mechanism of plant-derived bioactive peptides on inflammation is shown in Figure 1. The inflammatory factors, represented by interleukin-1β (IL-1β), interleukin-6 (IL-6) and tumor necrosis factor-α (TNF-α), are ultimately responsible for the inflammation in inflammatory bowel disease (IBD) [17], obesity and other inflammation-related diseases [18].

### 2.1. Plant-Derived Bioactive Peptides Regulate Single Inflammatory Signaling Pathways and Their Inflammatory Factors

#### 2.1.1. MAPK Pathways

Plant-derived bioactive peptides can mediate inflammation by regulating MAPK pathways. MAPK is a class of intracellular serine/threonine protein kinases that cause target protein phosphorylation with serine and threonine [19]. MAPK pathways include the three branch pathways of extracellular regulated protein kinases (ERK), c-Jun N-terminal kinase (JNK) and p38 MAPK (p38), which play important roles in stress responses, such as inflammation and apoptosis. Significant phosphorylation of the three protein kinases occurs in the cells following the stimulation by inducible factors. Plant-derived bioactive peptides can attenuate the inflammatory response, primarily by inhibiting ERK and p38 phosphorylation. In one study, the phosphorylation of ERK, JNK and p38 was inhibited by millet bran peptides [20]. The protein phosphorylation level of FGPK group (millet bran peptides) was not significantly different from that of the normal group. In another study, wheat peptides blocked MAPK signaling by downregulating the protein phosphorylation levels of ERK and p38 [21]. Compared to the cimetidine group, wheat peptides showed no significant difference in alleviating alcohol-induced gastric mucosal damage in mice. A comparison of different inflammation models revealed that plant-derived bioactive peptides were more effective at inhibiting the phosphorylation of threonine in ERK proteins via in vitro and in vivo experiments [20,21].

#### 2.1.2. NF-κB Pathway

Most inflammatory diseases, including allergies and endothelial injury, are associated with the disruption of the NF-κB pathway [22]. Plant-derived bioactive peptides have been shown to suppress inflammation by affecting key protein phosphorylation in the NF-κB pathway and regulating the phosphorylation cascade response. The main transcription factors in the NF-κB pathway are p50 and p65. Under normal physiological conditions, NF-κB’s binding to its inhibitory protein IκB is inactivated in the cytoplasm. However, pro-inflammatory cytokines activate the IKK complex, which phosphorylates the IκB protein, leading to autoubiquitination and proteasome degradation. The released NF-κB rapidly translocates from cytoplasm to nucleus, binding to nucleotropic DNA and regulating the expression of inflammatory factors.

The modifiable sites in the NF-κB signaling pathway have been explored in numerous studies to optimize regulation. Transcription factor p65 is a major site affected by plant-derived bioactive peptides. Several studies have discovered that plant-derived bioactive peptides can reduce the release of inflammatory factors by inhibiting p65 translocation, which are isolated from cereals [23], nuts [24] and other plant sources. Plant-derived bioactive peptides inhibited serine phosphorylation in p65 protein mainly by reducing the IKK complex and IκB protein phosphorylation. The anti-inflammatory effect of corn silk peptides was achieved through interaction with IKKβ [25], which inhibited IKKβ phosphorylation two-fold more than LPS in BALB/c mice. Meanwhile, it has been found that rice bran peptides inhibited the NF-κB signaling pathway by decreasing IκB protein phosphorylation [20].

The NF-κB pathway is the most straightforward pathway in the regulation of inflammation, having direct regulation of inducible nitric oxide synthase (iNOS), IL-1β, IL-6 and TNF-α. Inflammatory factors are critical markers of many diseases, and their changes can predict disease onset and characterize the recovery status of the organism. The monitoring and targeting of inflammatory factors are crucial for the treatment of inflammation and chronic diseases. Shi et al. found that bean peptides inhibited inflammation by regulating the NF-κB signaling pathway through transcriptomic analysis. The inhibitory effect on the specific regulator IL-1β was fourfold higher than that on non-specific inflammatory factors [26].

The mechanisms of MAPK and NF-κB pathway regulation by plant-derived bioactive peptides were clarified by analyzing the cascade response of a single pathway, and ERK and p65 proteins were found to be the specific sites of action. Moreover, changes in the inflammatory factors IL-1β, IL-6, and TNF-α can significantly characterize the effects of PAPs.

The impacts of PAPs on the janus kinase/signal transducer and activator of transcription (Jak/Stat) pathway have been less often investigated in recent years, since they have less literature behind them and require longer experimental times. However, the Jak/Stat pathway remains to be explored in depth as a critical pathway for chronic inflammation in addition to NF-κB and MAPK pathways [27].

### 2.2. Plant-Derived Bioactive Peptides Regulate Multiple Inflammatory Signaling Pathways and Their Inflammatory Factors

Inflammatory signaling pathways are complicated network structures with complex interactions among targets, and therefore, the simultaneous consideration of multiple pathways is required. The PI3K/Akt pathway, a crossover pathway of multiple signaling pathways, can regulate both the MAPK and NF-κB pathways that indirectly inhibit inflammation [28]. Akt protein phosphorylation can act on the ERK of the MAPK pathways, and the p38 and IKKβ of the NF-κB pathway. Kyoto Encyclopedia of Genes and Genomes (KEGG) class analysis has shown that lupin peptides exert anti-inflammatory effects by regulating p38 in PI3K/Akt and MAPK pathways and inhibit the secretion of inflammatory factors, such as TNF-α, IL-1, IL-6 and monocyte chemotactic protein 1 (MCP-1) [29]. An investigation has shown that corn oligopeptides reduced IKKβ phosphorylation in the NF-κB pathway in vivo by decreasing the phosphorylation of the Akt protein [30]. The effect of corn oligopeptides was significant compared to the LPS-induced group.

The MAPK and NF-κB pathways are prominent pathways to inflammation. The MAPK pathways, which are upstream of the NF-κB pathway, reduce NF-κB nuclear translocation by inhibiting the dissociation of IκB from NF-κB. Millet peptides were found to reduce IκB phosphorylation and inhibit the secretion of downstream inflammatory factors IL-6, TNF-α, IL-β and nitric oxide (NO) by inhibiting p-p38 and p-JNK production [31]. Walnut peptides inhibited p38 phosphorylation and p-IκB production, thereby inhibiting the nuclear translocation of p65 and reducing the secretion of downstream inflammatory factors TNF-α, IL-6, IL-1β, iNOS and cyclooxygenase (COX-2) in in vitro cellular assays [32]. Compared to the LPS-induced group, the anti-inflammatory effect of walnut peptides WEKPPVSH was significant.

Comprehensive analysis of these inter-pathway interactions has revealed that p38 is an important site linking the three inflammatory pathways of PI3K/Akt, MAPK and NF-κB, and is a major regulatory site for PAPs. Plant-derived bioactive peptides inhibit pathway activity by reducing the key protease phosphorylation. However, the exact mode of this intervention is still unknown. The intervention pathways may include the inhibition of kinase activity that interacts with the phosphate group, activation of phosphatase activity that removes the phosphate group or binding to the phosphate group to reduce kinase phosphorylation. Clarification is required in future studies of the intervention pathways and protein changes of PAPs on the targets and specific regulatory sites of interactions among key inflammatory pathways, which will assist in achieving focused regulation.

## 3. Factors Affecting the Anti-Inflammatory Effects of Plant-Derived Bioactive Peptides

The functions of plant-derived bioactive peptides are related to their structures. The analysis of their relationships can accurately predict their anti-inflammatory effectivity. Molecular weight, amino acid composition and location are the main factors affecting the anti-inflammatory abilities of plant-derived bioactive peptides.

### 3.1. Effect of Molecular Weight on the Anti-Inflammatory Properties of Plant-Derived Bioactive Peptides

The anti-inflammatory effect of a plant-derived bioactive peptide is influenced by its molecular weight. Peptides with low molecular weights (less than 1 kDa) tend to have higher levels of anti-inflammatory activity. Various studies have suggested that plant-derived bioactive peptides with molecular weights of approximately 500 Da (composed of 2-6 amino acids) have the strongest anti-inflammatory activity [33]. In one study, bee pollen hydrolysates were divided into five fractions based on molecular weight, and the fraction with the smallest molecular weight (MW < 0.65 kDa) was demonstrated to have the highest NO inhibitory activity [34]. Anti-inflammatory peptides were obtained from spirulina, and it was found that the peptides LDAVNR (686 Da) and MMLDF (655 Da) with molecular weights around 500 Da had the strongest inhibitory activity against IL-8 produced by endothelial cells EA.hy926 [35]. Although various low-molecular-weight peptides with anti-inflammatory activity have been validated via in vitro models, anti-inflammatory effects have yet to be validated in vivo. Endopeptidases in the stomach, intestinal lumen and brush border constitute the main chemical barriers to peptide bioavailability, which leads to disruption of the structures and anti-inflammatory activities of plant-derived bioactive peptides. Low-molecular-weight peptides have low enzymatic recognition and few cleavage sites. Reduced enzymatic cleavage facilitates their entry into the bloodstream as intact structures to reach their target organs and exert anti-inflammatory effects [36]. Furthermore, there are specific transport modes for low-molecular-weight peptides (Figure 2) among the four peptide transport modes present in the intestine (PepT1 transporters, cellular bypass, cytokinesis and passive diffusion). Dipeptides and tripeptides can be directly absorbed via PepT1 transporters (which have a high transport capacity). Soybean tripeptide VPY has been shown to achieve anti-inflammatory effects via PepT1 transporters [37]. Meanwhile, low-molecular-weight peptides (composed of 2–8 amino acids) can be transported via tight junction proteins in the cellular bypass pathway. Neither of these modes of transport require energy consumption; they avoid enzymatic digestion and retain the complete anti-inflammatory active fragment [38]. Moreover, low-molecular-weight peptides improve absorption by reducing endopeptidase enzymatic digestion (pepsin, trypsin, chymotrypsin, etc.), with specific and efficient transport modes, consequently increasing the potential of plant-derived bioactive peptides to reach the blood and target organs intact to exert their anti-inflammatory effects [39].

### 3.2. Effect of Amino Acid Composition on the Anti-Inflammatory Properties of Low-Molecular-Weight Plant-Derived Bioactive Peptides

Besides molecular weight, the functions of peptides are closely related to amino acid composition. In low-molecular-weight peptides, the folding of the spatial structures is restricted, thereby fully exposing the amino acids and increasing their anti-inflammatory effects. We summarize 39 recently reported, plant-derived bioactive peptides with anti-inflammatory activity and analyze their key amino acid compositions (Table 1).

#### 3.2.1. Hydrophobic Amino Acids

Regardless of chain length, most peptides with anti-inflammatory effects contain hydrophobic amino acids (Figure 3A). As presented in Table 1, 89.7% of the plant-derived bioactive peptides summarized herein contain at least one hydrophobic amino acid, and 53.8% include two or more hydrophobic amino acid structures. Leucine is the main amino acid, and the presence of tryptophan and phenylalanine further strengthens its anti-inflammatory activity. It was also found that PAPs generally have repetitive leucine and isoleucine structures. Seven of the twenty-three peptides with two or more hydrophobic amino acids have similar structures; these are mostly located in the middle of the peptides. Highly hydrophobic plant-derived bioactive peptides exert anti-inflammatory effects by enhancing the binding to cell membranes and membrane depolarization, thereby disrupting the cascade pathway of inflammation. Leucine and isoleucine can act specifically on PI3K, Akt kinases in the PI3K/Akt signaling pathway and ERK kinases in the downstream MAPK pathways [50]. They alleviate the damage of inflammatory factors by reducing the phosphorylation of kinases in the pathway and converting macrophages from the M1 type to M2 [51]. In addition, highly hydrophobic plant-derived bioactive peptides can prevent lipopolysaccharide (LPS)-stimulated inflammatory responses by binding to LPS to form peptide-lipopolysaccharide complexes and scavenging LPS by inducing cell membrane charge reversal [52].

#### 3.2.2. Positively Charged Amino Acids

Positively charged amino acids, such as lysine, arginine and histidine, are important for enhancing the anti-inflammatory activity of plant-derived bioactive peptides (Figure 3B). In particular, lysine has been found in most anti-inflammatory peptides from different plant sources. The oligopeptides KLRSRNLLHPT and TNGRHSAKKH obtained from bee pollen were shown to inhibit the expression of COX-2, iNOS, IL-6 and TNF-α in RAW264.7 macrophages [34]. Green tea peptides (LAEQAER, VECTIPK, DAYVGDEAQSK and MASLALK) were discovered to have the ability to reduce iNOS and TNF-α in diabetic mice, and three of these four peptide fragments contained lysine [45]. Moreover, lysine can specifically regulate the phosphorylation of kinase ERK and the nuclear translocation of transcription factor NF-κB in the MAPK signaling pathways [53]. Furthermore, arginine has also been reported in numerous PAPs, such as bee pollen [34], walnuts [32] and rice bran [46], and it is speculated that the presence of arginine is likely to be associated with anti-inflammatory activity. Arginine specifically reduces p38 and ERK kinases phosphorylation in the MAPK pathways. Meanwhile, it also inhibits the expression of TLR4 receptors and reduces the nuclear translocation of the transcription factor p65 by inhibiting IκB kinase phosphorylation in the NF-κB pathway [54]. In addition to the cascade pathways that specifically regulate the inflammatory response, the presence of charged amino acids increases the effect of peptide absorption in the intestine. Ding et al. designed bioactive peptide sequences with charged amino acids at the N- and C-terminals. The results demonstrated that positively-charged, basic amino acids increase transport permeability through the monolayer membrane of Caco-2 cells, enabling the absorbed plant-derived bioactive peptides to enter into the circulation of body fluids or to reach target organs, thereby increasing anti-inflammatory function [55].

#### 3.2.3. Specific Amino Acids

Specific amino acids, represented by glycine and glutamine, can increase the anti-inflammatory activity of plant-derived bioactive peptides (Figure 3C). Among the four peptides obtained from the sunflower (YFVP, SGRDP, MVWGP and TGSYTEGWS), MVWGP and TGSYTEGWS reportedly have the strongest ability to inhibit IL-1β secretion, which is thought to be due to the presence of glycine, which enhances anti-inflammatory properties [48]. Glycine and the low-molecular-weight peptides that contain glycine were found to have high affinity for calcium binding and to interfere with Ca^2+^ signaling [56]. Both affected intracellular Ca^2+^ levels through the glycine-gated chloride channels and exerted anti-inflammatory effects by modulating the NF-κB signaling pathway [57]. Of the five anti-inflammatory peptides obtained from maize (QLPY, EYPSIQ, LTDPAAS, LPVGPQ and LLPSSQ), four were found to contain glutamine. The corn peptides reduced CCl_4_-induced liver fibrosis in mice by inhibiting the mRNA expression of IL-6 and TNF-α [30]. Furthermore, the anti-inflammatory peptides ALLLQAVQSQYEEK obtained from rice bran [41] and IQDKEGIPPDQQR obtained from lupin [29] have at least one glutamine residue and both have been found to reduce the secretion of inflammatory factors such as TNF-α, IL-1β and IL-6 in RAW264.7 cells. In addition, studies have shown that glutamine contributes to the anti-inflammatory role played by peptides in the regulation of the cascade responses of inflammatory pathways. Glutamine downregulated inflammatory factors IL-1β, IL-8, TNF-α and IL-6, and attenuated LPS-induced lung injury in mice by modulating ERK, JNK and p38 in the MAPK pathways [58].

### 3.3. Effects of Amino Acid Positions in Plant-Derived Bioactive Peptides on Their Anti-Inflammatory Properties

In addition to the types of amino acids, their positions related to overall function (Figure 3D). In the summary presented in Section 3.2, it was revealed that a large number of amino acids with anti-inflammatory effects tend to be located at the ends of the peptide chains. Peptides with hydrophobic amino acids that are present at the N-terminus tend to have strong anti-inflammatory activity. These include the walnut tripeptide LPF [42]; the oligopeptide IALLIPF obtained from millet [31]; the corn oligopeptides LTDPAAS, LPVGPQ and LLPSSQ [30]; and the hemp seed oligopeptides WVSPLAGRT and IGFLIIWV [40]. The hydrophobic amino acids at the N-terminus of the peptide chain confer excellent anti-inflammatory effects on peptides, which reduce inflammation by inhibiting the cascade responses of key inflammatory signaling pathways and the expression of downstream inflammatory factors. In contrast, the charged amino acids in PAPs are often located at their C-terminal ends, as seen in the green tea peptides LAEQAER, VECTIPK, DAYVGDEAQSK and MASLALK [45]; and the rice bran peptides VLER, WVGK, VALVR, LFGK and FGPK [20], in which the key amino acids lysine and arginine are located at the C-terminal ends of the carbon chains. The positions of these amino acid enhance the transport of plant-derived bioactive peptides across the cell membrane, thereby achieving increased anti-inflammatory activity. The end of the peptide chain is a unique reactive site at which amino acids are more exposed compared to the middle of the chain [59], indicating that the nature of terminal amino acids plays a significant role in the anti-inflammatory properties of peptides. However, univariate experiments regarding key amino acids located at the end and middle of the peptide chain are difficult and have received little experimental support. Hence, the effect of the locations of key amino acids in PAPs requires further investigation.

## 4. Preparation of Plant-Derived Anti-Inflammatory Peptides

Information on peptides can be identified more rapidly and accurately with the development of histological technologies (proteomics, peptidomics) [60]. The anti-inflammatory peptides are obtained by first identifying the plant source. The protein hydrolysate is obtained by a hydrolysis method. The protein hydrolysate is then isolated and purified, and then separated to obtain plant-derived bioactive peptides. Finally, the anti-inflammatory properties of the peptides are verified by in vivo and in vitro experiments.

### 4.1. Preparation of Plant-Derived Anti-Inflammatory Peptides 

#### 4.1.1. Chemical Hydrolysis

Chemical hydrolysis is the breaking of peptide chains in proteins using chemical reagents at a certain temperature and pH, including acid hydrolysis and alkaline hydrolysis [61]. Acid hydrolysis has the advantage that the hydrolysis is complete. However, tryptophan and tyrosine, which have high anti-inflammatory activity, are almost completely destroyed. Alkaline hydrolysis produces a spin isomerization effect and changes the structures of lysine and arginine, which have anti-inflammatory effects. Chemical hydrolysis is used less frequently because it requires large amounts of chemicals and tends to pollute the environment.

#### 4.1.2. Microbial Fermentation

Microbial fermentation is the breakdown of proteins into peptides using proteases produced by the microbial fermentation process [62]. It not only releases peptides, but also enhances the nutrition of the food through other means, such as the presence of prebiotics and probiotics. The fermented soybean peptides reduce IL-1β, IL-6, TNF-α and COX-2 in mouse serum, alleviating IBD [63]. However, microbial fermentation has limitations, such as not being easily controlled and being susceptible to microbial contamination [64], so its application is currently limited.

#### 4.1.3. Enzymatic Hydrolysis

Enzymatic hydrolysis has the advantages of not producing toxic substances, not destroying amino acids and high catalytic efficiency, which is the main method adopted for hydrolyzing proteins in the food industry [65]. The enzyme can be selected and the peptide structure can be predicted based on the specificity of the enzyme, as different proteases have different cleavage sites.

In recent years, new food processing technologies for protein structural modification and enzymatic hydrolysis are being developed to obtain more biologically active, higher yielding and structurally well-defined peptides.

Subcritical hydrolysis has received widespread attention as a new green technology for the release of bioactive peptides [66]. Subcritical hydrolysis has shown flexibility and controllability, though the specificity of protein cleavage is unclear [67]. Researchers have found that microwave technology for peptide vaccines preparation has high purity and low contamination properties [68]. Microwave-assisted hydrolysis has been shown to improve the biological activity of chia seed peptides [69]. In the future, microwave-assisted hydrolysis could be applied to the production of PAPs. Ultrasonic treatment offers a new technique for studying plant-derived bioactive peptides. It was reported that ultrasonic pretreatment could enhance the functionality of sweet potato protein hydrolysates [70].

### 4.2. Influence of Enzyme Type in Enzymatic Processing on the Anti-Inflammatory Activity of Plant-Derived Anti-Inflammatory Peptides

The main determining factor in the processing of PAPs via enzymatic method is the choice of enzymes. Different proteases have different enzymatic properties. Specific enzymes can be selected in the processing of PAPs with strong anti-inflammatory functions according to the structure-activity relationships of peptides, as summarized in the previous section. Hydrolysis methods such as chemical hydrolysis and microbial fermentation, are not able to obtain specific peptide structures.

#### 4.2.1. Single Enzyme Digestion

Proteases typically target specific peptide bonds in their protein structures. Alcalase specifically breaks the peptide bonds between hydrophobic amino acids and other amino acids, which is able to expose more hydrophobic amino acids compared to other proteases [71]. The presence and numbers of hydrophobic amino acids are important factors that influence the anti-inflammatory activity of plant-derived bioactive peptides. Diao et al. utilized alcalase, trypsin, neutrase and flavourzyme to enzymatically digest mung bean proteins. The results indicated that the mung bean protein hydrolysate (MBPH) has the strongest anti-inflammatory activity in the presence of alcalase [72]. Compared to the LPS-alone group, MBPH significantly reduced NO, iNOS, IL-6 and IL-1β secretion levels by 52.6%, 53.2%, 48.4% and 49.7%, respectively, in LPS-induced macrophages.

Papain and trypsin can specifically cleave alkaline amino acid side peptide bonds and enhance the anti-inflammatory activity of active peptides by exposing positively charged alkaline amino acids [73,74]. Udenigwe et al. investigated the anti-inflammatory effects of flaxseed protein hydrolysates obtained using different proteases, and found that the proteolytic obtained using papain had the greatest ability to inhibit NO production in RAW264.7. Its enzymatic digestive activity was superior to those of the remaining six proteases (cellulase, thermolysin, alcalase, ficin, pepsin and trypsin) [75]. Feng et al. adopted trypsin to hydrolyze selenium-rich brown rice proteins, and the resulting hydrolysates inhibited both mRNA and the protein expression of iNOS and COX-2 [41]. The selection of specific proteases can increase the anti-inflammatory properties of plant-derived bioactive peptides under the same conditions, justifying the use of enzymatic methods to obtain anti-inflammatory peptides.

#### 4.2.2. Compound Enzyme Enzymatic Digestion

A combination of multiple proteases tends to exhibit better enzymatic effects than single enzymes. More low-molecular-weight peptide fragments with the properties of hydrophobic amino acids or positively charged, basic amino acids can be obtained through the binding of non-specific and specific proteases, leading to enhanced anti-inflammatory activity. Millán-Linares et al. explored the hydrolysis of lupin protein by lysozyme, alcalase complex and single alcalase. The results revealed that the lupine protein hydrolysates obtained by complex enzyme hydrolysis had stronger anti-inflammatory effects than those obtained by single enzyme hydrolysis [76]. Since each enzyme has a different site of action, working synergistically on different peptide bonds of proteins, there is fuller exposure of specific anti-inflammatory amino acids and structures in the hydrolysis products, giving them lower molecular weights [77]. Therefore, complex enzymes are usually used for protein enzymatic hydrolysis.

Various enzymatic solutions are currently used to obtain plant-derived bioactive peptides of low molecular weight and with specific amino acid compositions by analyzing the properties of different proteases. Thus, the design and optimization of enzymatic protocols are significant in the research and application of PAPs.

### 4.3. Isolation and Purification of Plant-Derived Protein Hydrolysates

From a hydrolysis perspective, protein hydrolysates are mixtures of unhydrolyzed proteins, peptide chains of different lengths and free amino acids and charges. From a by-product utilization point of view, numerous processing by-products are still of great research and application value, but have high impurity contents. Separation and purification of hydrolysis products or process by-products are required to accurately assess the anti-inflammatory efficacy and identify the structures of bioactive peptides, in order to maximize their anti-inflammatory effects.

Separation and purification are often carried out by ultrafiltration, which is used to obtain peptides of different molecular weights, depending on the size of the membrane [78]. Sandoval-Sicairos et al. used ultrafiltration membranes of 10 and 3 kDa to separate the germinated amaranth protein hydrolysates into three fractions and to explore the anti-inflammatory activity of each fraction [79]. However, further separation and purification of the enzymatic product are required, as with the ultrafiltration technique it is difficult to obtain high purity peptides; the molecular weight cut-off is inaccurate and the filter membrane is prone to blockage. Chromatography is a physicochemical method of separation and analysis. It uses different forces on the hydrolysate in the stationary and mobile phases to separate the components to be analyzed, and then analyses them sequentially [80]. Chromatography is characterized by high selectivity, high separation efficiency and fast analysis, but it is susceptible to pollution and influenced by factors such as temperature and the nature of the mobile phase. Vo et al. used anion exchange chromatography and reverse-phase high performance liquid chromatography for the separation and purification of Spirulina protein hydrolysates [35]. The hydrolysates were divided into four fractions using high performance liquid chromatography (HPLC), and the anti-inflammatory activity of the different fractions was assessed.

In summary, single methods of separation and purification can be limited by equipment. Multiple techniques are often used in combination to achieve better separation of peptide mixtures to obtain high-purity peptides. Gao et al. obtained and validated the anti-inflammatory activity of the walnut peptide WEKPPVSH by sequential purification of walnut protein hydrolysate through a Sephadex G-25 gel column, a Sephadex G-15 column and an RP-HPLC C18 column [32].

Protein hydrolysates can be divided by molecular weight after separation, whereas lower-molecular-weight protein hydrolysates tend to have better anti-inflammatory activity and can be artificially selected for further studies. The protein hydrolysates obtained after purification are purer, removing impurities and unhydrolyzed macromolecular proteins from the mixture, resulting in a better anti-inflammatory effect.

The separation and purification of peptide mixtures are well established under laboratory conditions. However, the time required to go from enzymatic hydrolysis to isolation and purification is long, and the equipment is expensive. More exploration and process optimization are needed to improve the protocol for its widespread application in the production of industrial anti-inflammatory bioactive peptides.

### 4.4. Identification of Peptides from Plant-Derived Bioactive Peptides

After isolation and purification, high purity peptide fractions were obtained. It is necessary to identify the peptide fragments in order to characterize the structures of plant-derived anti-inflammatory peptides, including peptide sequence analysis and the application of mass spectrometry (MS) techniques [81]. MS has been widely used in peptide sequence identification due to its high efficiency, sensitivity and reproducibility. The mass-to-charge ratio (*m*/*z*) of each peptide is obtained using MS, and the amino acid sequence can be determined by accessing an online protein database. Common databases include BIOPEP (http://www.uwm.edu.pI/biochemia, accessed on 23 February 2022), UniProt (http://www.uniprot.org/, accessed on 28 May 2022), etc.

## 5. Limitations of Plant-Derived Anti-Inflammatory Peptides

Research on PAPs is maturing. Figure 4 shows the complete line of research from processing, to in vitro and in vivo activity validation, to the mechanisms of investigation. However, few PAPs products have been identified for clinical application and commercialization, and multiple factors limit their development.

### 5.1. Difficulties in Determining the Enzymatic Solutions of Plant-Derived Anti-Inflammatory Peptides

The amino acid composition and structural characteristics of highly active anti-inflammatory peptides are revealed according to their structure-activity relationships. However, enzymatic methods to obtain high percentages of PAPs in a targeted manner are lacking. There is little information on the changes in peptide structure during enzymatic hydrolysis of proteins or in the end products. The peptides obtained by enzymatic digestion with different enzymes are different in structure. It is not possible to increase the proportions of hydrophobic amino acids, positively charged amino acids and specific amino acids in peptides in a targeted way. The type of enzyme, the ratio of compounds and the sequence of enzymatic digestion are all key factors affecting the preparation of PAPs and require considerable experimentation to explore. Conditions such as pH, temperature and time also need to be optimized.

### 5.2. Low Yield of Plant-Derived Anti-Inflammatory Peptide Processing

Low molecular weight and high purity are two important characteristics of PAPs. However, enzymatic products are all mixtures, and after isolation and purification, the yields of PAPs tend to be low. The narrower the molecular weight control range, the higher the purity and the lower the yield. Much of the current research has focused on the functions and mechanisms of PAPs. The issue of increasing the yield of PAPs has been neglected, but it is the key to the application of PAPs. High yields and high proportions of PAPs in the target molecular weight range can be increased by controlling the enzymatic hydrolysis conditions.

### 5.3. The Lack of Clinical Studies on Plant-Derived Anti-Inflammatory Peptides

Research on plant-derived bioactive peptides has exploded recently. However, the lack of clinical data limits the use of PAPs. This is because data from in vitro cell experiments and in vivo animal experiments are difficult to convert into clinical data [82]. The bioavailability of PAPs and the processes of metabolism and attenuation in vivo are still unknown. The dosing and efficiency of PAPs compared to pharmaceutical drugs need to be investigated. In future experiments, the safety and pharmacokinetics of PAPs should be examined. The bioavailability of PAPs can be improved by using enzyme inhibitors or microencapsulation to reduce enzymatic cleavage, which are then widely distributed throughout the body. Longer tests following nutritional interventions and exploration of potential mechanisms are also needed.

## 6. Conclusions and Prospects

Plants are promising sources of anti-inflammatory peptides. In this paper, the specific mechanisms of PAPs were analyzed, and the cascade reactions in a single pathway and interactions between multiple pathways were compared, with a focus on the protein sites specifically regulated by PAPs and the changes in downstream inflammatory factors. Important conclusions regarding the structure-activity relationships of PAPs were obtained, in which the presence and locations of low-molecular-weight and specific amino acids were found to be key to their anti-inflammatory effects. The effects of the structure-activity relationships were also demonstrated via transport absorption and cascade regulation in inflammatory pathways, and the selection of enzymes in the processing of PAPs was recommended based on the confirmation of the structure–activity relationship. Enzymatic digestion is based on a basic combination of specific and non-specific enzymes, which increases the exposure of specific amino acids and structures in peptides through certain enzyme changes, leading to increased anti-inflammatory activity. However, the specific processes and the structure-activity relationships of PAPs are still unknown, and enzymatic processing presents difficulties in obtaining amino acid compositions in a targeted manner and with low yields. Moreover, most studies on PAPs are in vitro cellular experiments; and the lack of animal models, human clinical trials and pharmacokinetics limits the applications of PAPs. In future, these challenges should be explored in depth and overcome so as to achieve the wider application of plant-derived bioactive peptides in the prevention and treatment of inflammatory-related chronic diseases.

## Figures and Tables

**Figure 1 foods-11-02361-f001:**
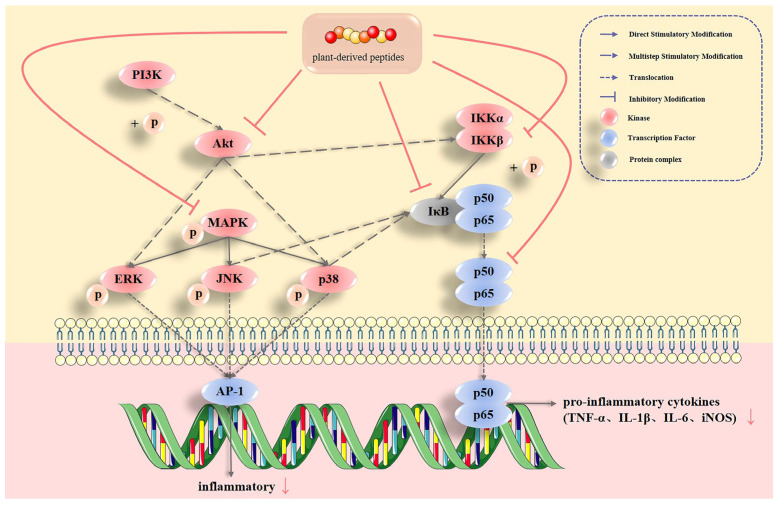
Regulation of inflammatory signaling pathways by plant-derived bioactive peptides. (PI3K: phosphatidylinositol 3-kinase; Akt: protein kinase B; MAPK: mitogen-activated protein kinase; ERK: extracellular regulated protein kinases; JNK: c-Jun N-terminal kinase; p38: p38 MAPK; AP-1: activator protein 1; IKK: inhibitor of kappa B kinase; IκB: inhibitor of NF-κB; p: phosphorylation; TNF-α: tumor necrosis factor-α; IL-1β: interleukin-1β; IL-6: interleukin-6; iNOS: inducible nitric oxide synthase).

**Figure 2 foods-11-02361-f002:**
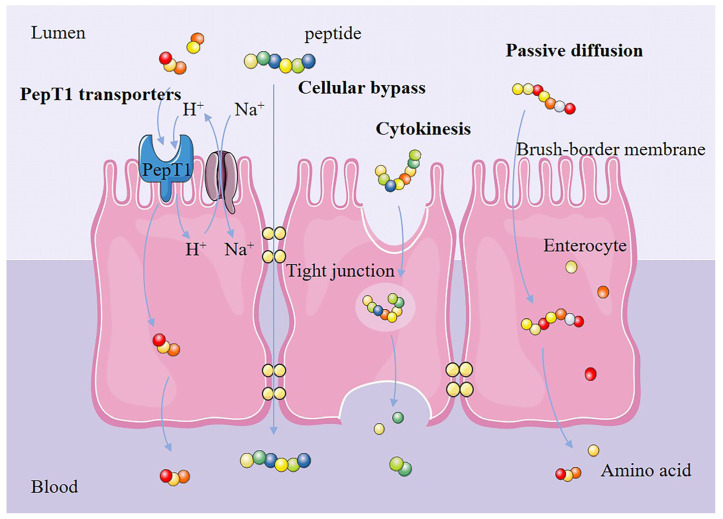
Transport mode of low-molecular-weight, plant-derived bioactive peptides in the intestine.

**Figure 3 foods-11-02361-f003:**
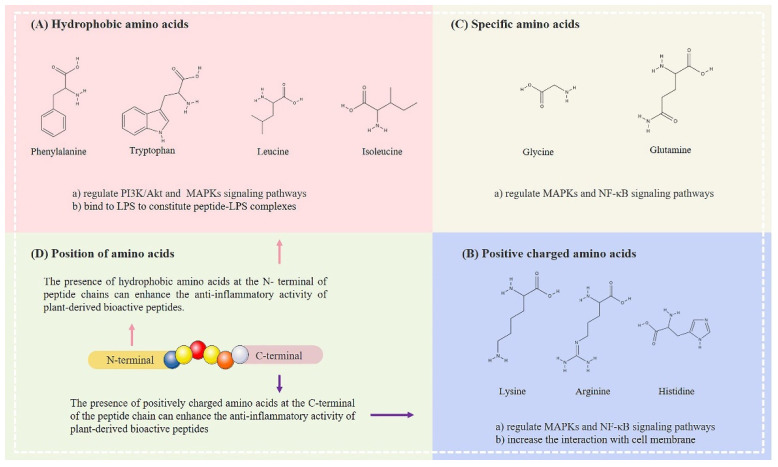
Effects of amino acid composition and position on the anti-inflammatory activity of low-molecular-weight bioactive peptides.

**Figure 4 foods-11-02361-f004:**
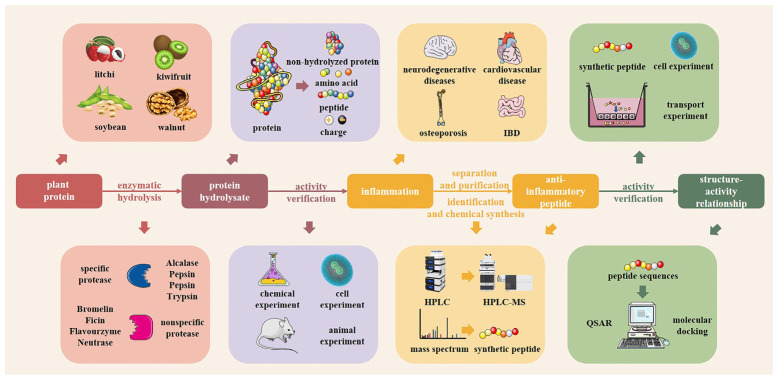
Processing, activity verification and mechanism investigation of plant-derived anti-inflammatory peptides.

**Table 1 foods-11-02361-t001:** Effects of amino acid composition on the anti-inflammatory properties of low-molecular-weight, plant-derived bioactive peptides (2020–2022).

Amino Acid Composition Characteristics	Peptide Sequences	Source	Models	Signaling Pathways	Inhibition of Pro-Inflammatory Factors	References
High hydrophobic amino acid content	WVSPLAGRT, IGFLIIWV	Hempseed	HepG2 cells	NF-κB	NO, iNOS	[40]
	VLER, WVGK, VVRP, VLLF, VALVR, LFGK, FGPK	Millet bran	RAW264.7 cells	MAPK; NF-κB	TNF-α, IL-1β, PGE2	[20]
	ALLLQAVQSQYEEK	Brown rice	RAW264.7 cells	MAPK; NF-κB	IL-6, IL-1β, TNF-α, iNOS, COX-2	[41]
	LPF	Walnut	RAW264.7 cells	NF-κB	iNOS, COX-2, TNF-α, NO	[42]
	NSPGPHDVALDQ, RMVLPEYELLYE	Chia seed	RAW264.7 cells	NF-κB	iNOS, NO, PGE2, TNF-α	[43]
	LPF, GVYY, APTLW	Walnut	BV-2 cells	-	TNF-α, IL-1β, IL-6	[44]
	PFLF, IALLIPF	Millet	RAW264.7 cells	MAPK; NF-κB	IL-6, TNF-α, NO, IL-β	[31]
High positively charged amino acids content	LAEQAER, VECTIPK, DAYVGDEAQSK, MASLALK	Green tea	HK-2 cells	NF-κB	iNOS, TNF-α	[45]
	WEKPPVSH	Walnut	BV-2 cells	MAPK; NF-κB	TNF-α, IL-6, IL-1β, iNOS, COX-2	[32]
	KLRSRNLLHPT, TNGRHSAKKH	Bee pollen	RAW264.7 cells	-	COX-2, IL-6, iNOS, TNF-α	[34]
	KHNRGDEF	Rice bran	D-gal-treated mice	NF-κB	-	[46]
	WSREEQEREE, ADIYTEEAGR	Walnut	UV- induced mice	NF-κB	IL-1β, IL-6	[47]
High specific amino acids content	QLPY, EYPSIQ, LTDPAAS, LPVGPQ, LLPSSQ	Corn	CCl_4_- induced mice	PI3K/Akt; NF-κB	-	[30]
	KQSESHFVDAQPEQQQR	Adzuki bean	RAW264.7 cells	NF-κB	IL-1, IL-6, TNF-α, MCP-1	[26]
	IQDKEGIPPDQQR	Lupin	RAW264.7 cells	MAPK	TNF-α, IL-1, IL-6, MCP-1	[29]
	YFVP, SGRDP, MVWGP, TGSYTEGWS	Sunflower	THP-1 cells	NF-κB	IL-1β	[48]
	YDWPGGRN	Wheat germ	RAW 264.7 cells	NF-κB	NO, IL-1β, IL-6, TNF-α	[49]

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
