# Peer review of "Anti-Inflammatory Function of Plant-Derived Bioactive Peptides: A Review"

_foods, 2022, doi:10.3390/foods11152361_

Round 1

Reviewer 1 Report

Well designed review paper with sound data that support the conclusions. The only correction suggested is that section 5.1 to 5.4 are very short. I suggest to expand the information or reorganize the information. 

Author Response

Dear Reviewer 1:

Thank you for your comments on our manuscript. We appreciate you for the time and effort invested in reviewing the previous version of the manuscript. The suggestions enabled us to improve our work. We have read and addressed the suggestions carefully, and the revised manuscript with all the modifications marked in red. The comments are reproduced, and our responses are provided directly below in red.

Responses to Reviewer 1’s comments

Point 1: Well designed review paper with sound data that support the conclusions. The only correction suggested is that section 5.1 to 5.4 are very short. I suggest to expand the information or reorganize the information.

Response 1: Thank you for your comments and support. Section 5.1 has been combined with relevant information to avoid forming a paragraph due to minimal content. As shown below:

“The impact of PAPs on the janus kinase/signal transducer and activator of transcription (Jak/Stat) pathway has been less investigated in recent years, since it contains fewer research bases and requires longer experimental time. However, the Jak/Stat pathway remains to be explored in depth as a critical pathway for chronic inflammation in addition to NF-κB and MAPK pathways.” This section is in 2.1, L147.

“Comprehensive analysis of these inter-pathway interactions has revealed that p38 is an important site linking the three inflammatory pathways of PI3K/Akt, MAPK, and NF-κB, and is a major regulatory site for PAPs. Plant-derived bioactive peptides inhibit pathway activity by reducing the key protease phosphorylation occurring. However, the exact mode of this intervention is still unknown. The intervention pathways may include the inhibition of kinase activity that interacts with the phosphate group, activation of phosphatase activity that removes the phosphate group, or binding to the phosphate group to reduce kinase phosphorylation. Clarification is required in future studies of the intervention pathways and protein changes of PAPs on the targets and specific regulatory sites of interactions among key inflammatory pathways, which will assist in achieving focused regulation.” This section is in 2.2, L177.

We have expanded the information for other sections. As shown below:

“5.1. Difficulties in determining the enzymatic solution of plant-derived anti-inflammatory peptides

The amino acid composition and structural characteristics of highly active anti-inflammatory peptides are revealed according to their structure-activity relationship. However, enzymatic methods to obtain a high percentage of PAPs in a targeted manner are lacking. There is little information on the changes in peptide structure during enzymatic hydrolysis of proteins or in the end products. The peptides obtained by enzymatic digestion with different enzymes are different in structure. It is not possible to increase the proportion of hydrophobic amino acids, positively charged amino acids and specific amino acids in peptides in a targeted way. The type of enzyme, the ratio of compounding and the sequence of enzymatic digestion are all key factors affecting the preparation of PAPs and require considerable experimentation to explore. Conditions such as pH, temperature and time also need to be optimised.

5.2 Low yield of plant-derived anti-inflammatory peptide processing

Low molecular weight and high purity are two important characteristics of PAPs. However, enzymatic products are all mixtures and, after isolation and purification, the yield of PAPs tends to be low. The narrower the molecular weight control range, the higher the purity and the lower the yield. Much of the current research has focused on the function and mechanism of PAPs. The issue of increasing the yield of PAPs has been neglected, but it is the key to the application of PAPs. High yields and high proportions of PAPs in the target molecular weight range can be increased by controlling the enzymatic hydrolysis conditions.

5.3 The lack of clinical studies on plant-derived anti-inflammatory peptides

Research on plant-derived bioactive peptides has exploded recently. However, the lack of clinical data limits the use of PAPs. This is because data from in vitro cell experiments and in vivo animal experiments are difficult to convert into clinical data [82]. The bioavailability of PAPs and the processes of metabolism and attenuation in vivo are still unknown. The dosing and efficiency of PAPs compared to pharmaceutical drugs need to be investigated. In future experiments, the safety and pharmacokinetics of PAPs should be examined. The bioavailability of PAPs can be improved by using enzyme inhibitors or microencapsulation to reduce enzymatic cleavage which is widely distributed throughout the body. Longer tests following nutritional interventions and exploration of potential mechanisms are also needed.”

We acknowledge and thank you for the professional comments and suggestions, which are valuable in improving our manuscript's quality. All the lines and pages indicated above appear in the revised manuscript.

Reviewer 2 Report

Improve the quality/resolution of the figures.

Figure 3 shows mistakes.:

A)Hydrophobic amino acids;

D) Specific amino acids;

D) Position of amino acids and

B) Positive charged amino acids.

Author Response

请参阅附件。

Reviewer 3 Report

L11- "2019" can it be replaced with the more recent update?

L14 - please provide an efficiency rate for those diseases in comparison with the modern/pharmaceutical drug, if possible. 

L25 - enzymatic processing is more suitable than preparation.

L28-38 - lack references, please include.

L52-54 - author mentioned plant-dervied bioactive peptides found in cereals, fruits, etc.,  and however, they are usually found in the protein form, isn't it? please check.

Introduction lack of processing or production of such bioactive compounds from plant-based proteins, please include some information about it.

L71 - Inflammation occurs in four stages... this sentence seems very short, please could you modify and connect the following sentence next to it? or list those four stages first and then explain them individually.

L82 - Please include a note section, and add expansion for all of those abbreviations given in figure 1, I understood the authors have already provided them in the text, but however, it is recommended to provide abbreviations under the legend of the illustrations and it could be more helpful for readers.

The listed plant-based bioactive peptides in section 2.1 are naturally present in this form, or they have been extracted? if so, this section and its subsection lack information about its origin and processing conditions used to produce, please revise and add the information, accordingly. 

MAPK pathways - the mode of action of the peptides on regulation MAPK is not clearly understandable, and the given information is limited, please fix it.

L101-103 - evidence for this finding?

Please include an illustration of the mode of action of the peptides against the NF-kB pathway.

A common problem in section 2 is, that the author provided a statement regarding the effect of peptides against those listed pathways, but not clearly established, example, in many places they mentioned, "inhibit inflammation" or "control chronic diseases" etc, but no reference given nor clear explanation is provided. 

I recommend authors include a section regarding the production of bioactive peptides, methods involved and yield, etc. I have noticed the enzyme method, there are other methods available also, please include them. Also, discuss a variety of protease enzymes and their efficacy in controlling inflammation using their end product. 

Please make sure, that every given abbreviation in the manuscript text contains an explanation as well, in L174 - the author mentioned NO for the first time, but did not provide the expansion, please check the manuscript thoroughly and fix it wherever possible. 

L194 - list some of the endopeptidase enzymes in brackets. 

L197 - provide a reference.

In many areas in the review paper, the author did not provide sufficient reference, some section has just 2 references and but the information is a lot, and probably not all of them is not directly conveyed by the author itself, those areas, please provide references of the sources. 

Please check the clarity of the figures, they are blurry upon expansion, difficult to see the given information in the figures, please fix it.

If the author includes the infographic of protein hydrolyzation by various proteases and their by-products would help this review more. I noticed figure 4, but the given information is very minimal. 

Section 5.1, should be combined into one or big paragraph based on the relevant information, for now, it is very minimal and not necessarily needed to have subsections. 

References are minimal in terms of a review paper, it is always recommended to have more than 70. Please add more references to the text, there are numerous places it requires references. 

Author Response

请参阅附件。

Round 2

Reviewer 3 Report

This paper is revised well, according to the comments, therefore it can be accepted for publications.